# Functional Mimicry of Eukaryotic Actin Assembly by Pathogen Effector Proteins

**DOI:** 10.3390/ijms231911606

**Published:** 2022-10-01

**Authors:** Saif S. Alqassim

**Affiliations:** College of Medicine, Mohammed Bin Rashid University of Medicine and Health Sciences, Building 14, Dubai Health Care City, Dubai P.O. Box 505055, United Arab Emirates; saif.alqassim@mbru.ac.ae

**Keywords:** cytoskeleton, actin, nucleation, elongation, mimicry, bacterial pathogen, virulence, effector protein

## Abstract

The actin cytoskeleton lies at the heart of many essential cellular processes. There are hundreds of proteins that cells use to control the size and shape of actin cytoskeletal networks. As such, various pathogens utilize different strategies to hijack the infected eukaryotic host actin dynamics for their benefit. These include the control of upstream signaling pathways that lead to actin assembly, control of eukaryotic actin assembly factors, encoding toxins that distort regular actin dynamics, or by encoding effectors that directly interact with and assemble actin filaments. The latter class of effectors is unique in that, quite often, they assemble actin in a straightforward manner using novel sequences, folds, and molecular mechanisms. The study of these mechanisms promises to provide major insights into the fundamental determinants of actin assembly, as well as a deeper understanding of host–pathogen interactions in general, and contribute to therapeutic development efforts targeting their respective pathogens. This review discusses mechanisms and highlights shared and unique features of actin assembly by pathogen effectors that directly bind and assemble actin, focusing on eukaryotic actin nucleator functional mimics *Rickettsia* Sca2 (formin mimic), *Burkholderia* BimA (Ena/VASP mimic), and *Vibrio* VopL (tandem WH2-motif mimic).

## 1. Introduction

Pathogens manipulate host cell biological processes to their advantage during infection. This is achieved through direct host–pathogen interactions: by encoding effectors or virulence factors (pathogen) that bind proteins inside an infected cell (host), thereby manipulating their cellular functions. A major target for these pathogens is the actin cytoskeleton [1,2,3,4], whose dynamics are fundamental to many essential cellular processes such as division, motility, endocytosis, and intracellular trafficking [5]. Pathogens have evolved strategies to hijack actin assembly to benefit various aspects of their life cycle such as invasion [6], motility [7], and cell-to-cell spread [8]. This includes control of Rho GTPase signaling upstream of actin assembly pathways [9], or hijacking and activation of the Arp2/3 complex [10]. Furthermore, some pathogens encode effectors or toxins that directly bind or modify actin to promote non-canonical actin:actin interactions, distorting regular actin dynamics as a result [11,12,13,14,15]. Some pathogens, however, encode effectors that directly interact with and polymerize actin, functionally mimicking eukaryotic actin assembly [16].

These effectors are of special interest for two major reasons. Firstly, they are the products of convergent evolution [17], i.e., they use novel sequences (making them difficult to identify using in silico approaches), folds, and mechanisms to polymerize actin, devoid of the regulatory sequence elements found in eukaryotic nucleators that keep them inactive or autoinhibited, and are often less complex as a result [18]. Cellular control of actin assembly has and continues to be the subject of intense research interest given how central actin is to essential cellular processes [19,20,21,22]. Molecular dissection of their mechanism of actin assembly offers an opportunity to gain significant insights into fundamental determinants for actin assembly [23,24,25,26]. Finally, their study would inform our understanding of host–pathogen interactions, pathogenicity mechanisms in general, and offer the potential of targeting their respective disease-causing pathogen with therapeutics.

Here, the current knowledge on the structure, function, and mechanism of actin polymerization by pathogenic effector proteins is reviewed, with an emphasis on pathogenic effectors that are direct functional mimics of eukaryotic nucleators: *Rickettsia* Sca2 (formin mimics), *Burkholderia* BimA (Ena/VASP mimics), and *Vibrio* VopL/F (tandem WH2-motif mimics). In addition, these are discussed in the context of their eukaryotic mimics, highlighting several shared and unique features.

## 2. Actin Assembly in Eukaryotes

Actin is a globular protein (G-actin) that can self-associate, or polymerize, into filaments (F-actin). The initial step of actin filament assembly, namely nucleation, is energetically unfavorable: it requires the formation of nuclei (dimers or trimers) that can seed the incorporation of additional actin monomers to form a filament [19,27]. Furthermore, most of the monomeric actin in cells is bound to the monomer-binding proteins profilin or thymosin-β4, which inhibit spontaneous nucleation [28]. This limitation is overcome by the use of specialized actin nucleation and elongation factors that can stabilize small actin oligomers (dimers, trimers, or tetramers) and catalyze the addition of actin subunits [24]. The major families of eukaryotic nucleators, discussed below, achieve this using different mechanisms.

### 2.1. Arp2/3

Arp2/3 complex (hereafter Arp2/3) consists of seven proteins, including two actin-related proteins (Arp2 and Arp3), and is involved in the formation of branched actin networks [29]. Arp2/3 by itself is inactive, requiring the action of nucleation promoting factors (NPFs). These NPFs have conserved C-terminal regions comprising a Pro-rich domain (PRD), WASP-homology 2 (WH2) motif(s), Central (C) and Acidic (A) regions that bind, respectively, profilin-actin, actin, and Arp2/3 itself. Arp2/3 activation, and subsequent actin nucleation/elongation, requires the binding of two NPFs which bring in two actin subunits, and a conformational change of Arp2 and Arp3 where they resemble a pseudo-actin dimer [22]. Despite the existence of Arps in prokaryotic genomes, there are no known pathogen effectors that mimic the eukaryotic Arp2/3 complex for actin assembly. This could be, perhaps, due to the complex activation of Arp2/3, cellular NPFs themselves being autoinhibited and under control of signaling pathways, and the abundance of Arp2/3 in eukaryotic cells targeted by pathogens. Instead, pathogens target eukaryotic Arp2/3 by encoding NPF mimics or upstream activators of NPFs. As these effectors have not been shown to directly assemble actin filaments and have been reviewed elsewhere [30,31], they will not be discussed further here.

### 2.2. Formins

Formins are a family of proteins that nucleate and elongate long unbranched filaments [32]. Although mammalian formins vary in their N-termini, most contain adjacent formin-homology 1 (FH1) and 2 (FH2) domains (Figure 1A). Formins exist in an autoinhibited state due to an intramolecular interaction. Upon activation, or rather, relief of autoinhibition by Rho GTPases, the doughnut-like FH2 domain dimer, with each ‘half’ of the doughnut being contributed by a single FH2 monomer, is able to mediate nucleation by encircling and stabilizing two actin monomers. Formins remain bound to the barbed end after nucleation and during elongation. Processive elongation, or addition of actin subunits to barbed ends bound by formins, can occur either by direct binding of actin monomers to the barbed end, or by delivery of actin monomers from profilin-actin bound to PRDs in the disordered FH1 domain. The probability a bound FH2 dimer is found in an ’open’ conformation that permits direct addition of actin monomers to the barbed end is termed the ‘gating factor’: the higher the gating factor (e.g., mammalian formin mDia1), the higher the probability that elongation can proceed by direct addition of actin monomers to the barbed-end-bound formin. On the other hand, formins with a low gating factor (e.g., yeast formin Cdc12) elongate almost entirely from profilin-actin bound to PRDs. Different mammalian formin isoforms vary in their gating factors. As a result, formin isoforms appear to be fine-tuned for processive elongation of actin subunits either directly or from the large profilin-actin pool [33,34,35].

### 2.3. Ena/VASP

Ena/VASP proteins are similar to formins in that they elongate, and in some cases also nucleate, long unbranched actin filaments [36]. In their C-terminal regions, mammalian Ena/VASP isoforms contain PRDs that can bind profilin-actin, a G-actin binding (GAB) motif, an F-actin binding (FAB) motif, followed by a coiled-coil (CC) region that mediates tetramerization (Figure 1C) [37]. Both the GAB and FAB display high sequence homology to the WH2 motif, a prominent actin-binding motif. These sequence elements are connected by flexible regions. As such, Ena/VASP proteins use a combination of oligomerization and actin-binding to catalyze the processive addition of actin subunits to multiple, usually bundled, actin filament barbed ends. Clustering of Ena/VASP, likely on membranes, also plays an enhancing role in Ena/VASP actin assembly [38,39,40,41,42].

### 2.4. Tandem Actin Monomer-Binding Proteins

Some actin nucleators, such as Spire (Figure 1E) [43], Cobl [44], and leiomodin [45], do not belong to any of the aforementioned eukaryotic nucleator families. They nucleate actin by utilizing a tandem repeat of WH2 motifs, a sequence of 17–20 amino acids capable of binding an actin monomer in its hydrophobic cleft. The WH2 motifs are typically connected by flexible linkers to allow for the binding of multiple actin monomers in an orientation that is polymerization-competent. Actin nucleation is driven by and dependent upon the number of and spacing between each WH2 motif [46,47,48]. Despite their ability to nucleate, they tend to synergize with other proteins for efficient nucleation in vivo, where their indirect oligomerization through their interactors tend to be important [49].

## 3. Actin Assembly by Pathogenic Effector Proteins

### 3.1. Rickettsia Sca2

*Rickettsiae* of the spotted fever group are Gram-negative intracellular pathogens that infect humans through arthropod vectors such as ticks and fleas [50,51], causing severe diseases such as rickettsiosis and spotted fever [52,53]. Inside the cytosol of infected human cells, the bacterium nucleates actin on its surface to employ an actin comet tail-based motility through two different proteins expressed on and anchored to its surface at different stages of infection: early post-infection, they use RickA, an NPF mimic, to activate Arp2/3 and form short and curved actin comet tails, whereas later post-infection, they use the autotransporter surface-cell antigen 2 (Sca2), forming long and unbranched actin comet tails. *Rickettsia* with a disrupted sca2 gene lose their ability to undergo actin-based motility, resulting in reduced virulence [54,55,56].

Similar to other autotransporters, or bacterial type Va secretion system proteins, surface exposure of Sca2 on the *Rickettsial* outer membrane first necessitates transport through the inner membrane using a signal sequence (residues 1–33), which is later cleaved. Then, the transmembrane C-terminal autotransporter domain (residues 1516–1795) forms a pore in the outer membrane through which the rest of the protein (passenger domain, residues 34–1515) is translocated and remains attached, exposed to the surface [57].

Other than two PRDs, Sca2 displays no sequence similarity or sequence elements and domain organization to any known eukaryotic actin nucleators (Figure 1B). Sca2 functionally mimics actin assembly by eukaryotic formins in that it nucleates and elongates unbranched filaments, processively associates with barbed ends, competes with capping protein for barbed-end binding, and relies on profilin-actin for efficient elongation [54]. Mechanistically, however, Sca2 is very different from formins in the manner with which it assembles actin filaments. Structural and biophysical experiments, discussed below, suggest that Sca2, through convergent evolution and structural mimicry, has ‘reproduced’ the feature of encircling two (or more) actin subunits for nucleation and barbed-end tracking [58,59].

Sca2 is an all-helical structure, predicted computationally by secondary structure prediction programs, and experimentally for large portions of the molecule: residues 34–400 (NRD, N-terminal repeat domain) by X-ray crystallography and residues 868–1515 by circular dichroism [58]. The structure of NRD revealed a novel fold comprised of helix–loop–helix repeats with an overall architecture that is strikingly similar to the curved crescent-like shape of a formin FH2 monomer [58]. It is hypothesized that the regions of Sca2 C-terminal to the NRD (residues 401–1355) adopt a fold similar to that observed in the NRD. Support for this idea comes from the fact that all autotransporters analyzed thus far adopt very simple folds dominated by local contacts with repetitive elements. This is likely a consequence of an autotransporter having to fold into its native conformation on the outer membrane without an energy source, which imposes severe constraints on its fold [60].

To nucleate and track the barbed end of actin, formins dimerize through their FH2 domains, each of which adopts a crescent-like fold, and forms half of a ‘doughnut’ that encircles two actin subunits [61,62]. Sca2 appears to reproduce this aspect of barbed-end tracking within a monomer. Co-expressed N- (residues 34–670) and C-terminal (residues 868–1515) halves of Sca2 interact and cooperate, reproducing the actin nucleation activity of the entire passenger domain (residues 34–1515) [58]. Analytical size exclusion experiments revealed that terminal truncations of the N-terminal half fail to interact with the C-terminal half, and vice versa, supporting the idea that the interaction between the two halves is end-to-end. The passenger domain (residues 34–1515) binds two actin subunits with nanomolar affinities, as demonstrated by isothermal titration calorimetry (ITC). The existence of putative WH2 motifs in the middle domain (residues 869–1060) was also ruled out by ITC experiments with monomeric actin. Instead, this region forms a contiguous globular actin-binding surface that contributes to the nucleation activity of the entire passenger domain (residues 34–1515). Taken together, these data suggest a model where Sca2 encircles two actin subunits in a ring-like manner, in a way analogous to a formin FH2 dimer, where each half contributes to binding a single actin monomer (Figure 2A) [59].

Utilizing its two PRDs, which contain three and two predicted profilin-actin binding sites, respectively, Sca2 has been shown to remain associated with the growing barbed end, elongating only in the presence of profilin-actin [58]. This suggests that Sca2, if analyzed similarly to a formin, displays a low apparent gating factor, as observed for some formins [63]. Though, in contrast with formins, which vary in the number and spacing between its PRDs [64], the two PRDs of Sca2 appear to synergize, as neither a single nor double Sca2 PRD mutant can elongate from profilin-actin [58].

In light of what-is-known about formins, several mechanistic questions remain [32,33]. With regards to barbed-end tracking, formins make use of the inherent flexibility within a single FH2 monomer and the ability of an FH2 monomer to undergo a conformational change with respect to the other to form contacts and accommodate an incoming actin subunit at the barbed end during elongation. With regards to elongation, formins elongate actin processively from the profilin-actin pool in cells, making use of the PRDs present on FH1 domains, which are N-terminally adjacent to the FH2 dimer, are inherently flexible, and are not part of the barbed-end tracking unit (FH2 dimer). Sca2 is monomeric and is mostly all-helical, with the only predicted flexibility or disordered regions in the two PRDs. The whole molecule is thought to form the doughnut-like circle around the barbed end, with its two PRDs 370 residues apart in the middle of the molecule, i.e., contiguous within the barbed-end tracking unit. How Sca2 tracks the barbed end and the conformational changes that are associated with accommodating and stabilizing an incoming actin subunit from profilin-actin are unclear and remain to be elucidated.

### 3.2. Burkholderia BimA

*Burkholderiae* of the pseudomallei group are Gram-negative intracellular pathogens that are known to cause the severe human disease known as melioidosis [65]. Similar to spotted fever group *Rickettsiae*, this pathogen employs an actin comet tail-based motility after vacuole escape inside infected human cells, forming long unbranched actin tails, through a trimeric autotransporter protein expressed on its surface localized at the pole of the bacterium: *Burkholderia* intracellular motility A (BimA) [66,67]. Mutation of BimA abolishes actin-based motility of *Burkholderia pseudomallei* in infected macrophage-like cells [66]. Actin-based motility is an important aspect of the life cycle of this *Burkholderia* species as it is required to mediate cell–cell fusion. What results is the formation of multi-nucleated giant cells, thereby allowing the pathogen to remain inside the cytosol, protected from extracellular immune system surveillance [68,69].

In vitro, BimA has been shown to function similar to eukaryotic Ena/VASP (Figure 2B) [70]. BimA is sufficient to nucleate actin filaments, as demonstrated by pyrene-actin assembly assays (fluorescence-based assay to follow bulk actin polymerization in solution) and visualized by epifluorescence microscopy. It displays processive elongation behavior: accelerating elongation ~3-fold compared to actin alone, and remains associated with growing barbed ends. Furthermore, similarly to Ena/VASP, it can elongate multiple filament barbed ends simultaneously, as observed by TIRF microscopy. Oligomerization, mediated by its C-terminal trimeric coiled-coil, is an absolute requirement for BimA: oligomerization mutants fail to nucleate actin, bind barbed ends, and compete with capping protein for barbed-end binding. BimA contains at least two predicted WH2 motifs, or GABs, which, when mutated, disrupt actin nucleation activity in vitro, and impact actin tail formation and cell–cell fusion in infected mammalian cells [70].

Autotransporters are either monomeric in nature, owing to a membrane-spanning translocator domain of ~300 amino acids, as is evident and experimentally demonstrated for Sca2, or trimeric, owing to a ~75 amino acid translocator domain [71]. BimA comes from the trimeric family of autotransporters [72], and has evolved to take full advantage of this oligomerization mechanism to mimic Ena/VASP actin assembly. BimA is trimeric, whereas Ena/VASP is tetrameric. Although the oligomeric state affects elongation acceleration and processivity, when Ena/VASP is clustered on a surface such as beads, it displays processive behavior irrespective of oligomerization state, i.e., even as a trimer [41,42]. The polar localization of BimA on the surface of *Burkholderia*, an important aspect of processive and directional actin-based motility, appears to mimic the clustering aspect of Ena/VASP on beads in vitro or on membranes in a physiological context [73].

At the sequence level, BimA displays elements that appear similar to the functional ‘business’ end of Ena/VASP: a PRD, followed by at least two putative WH2 motifs (Figure 1D). Several aspects of BimA mimicry of Ena/VASP are still unclear. For instance, three putative WH2 motifs are identified, and at least two of them are shown to play an important role in nucleation and binding monomeric actin through mutagenesis [70]. Whether these motifs are sufficient to bind monomeric actin in vitro, and whether they function as GABs similar to how Ena/VASP uses GABs, is yet to be tested. On a similar note, the mechanistic details and functional domain contribution to processivity and barbed-end tracking remain unclear: Does BimA use three (or more) GABs, and an unidentified FAB, to functionally mimic Ena/VASP elongation in terms of monomer capture and monomer to barbed-end transfer, or does it achieve this using a different mechanism? The role of the PRD and influence of profilin or profilin-actin, which constitutes a large majority of the cellular monomeric actin pool, on BimA are amongst the questions that warrant further investigation.

### 3.3. Vibrio VopL

*Vibrio* species are Gram-negative intracellular pathogens that cause various severe diseases such as gastrointestinal diseases and wound infection [74,75,76,77,78]. Inside human cells infected by *Vibrio*, various effectors are secreted into the cytosol which cause disruption of actin dynamics and play an important role in its infectious life cycle and intracellular survival [79,80]. Among those effectors is *Vibrio* outer protein L (VopL), which has the ability to directly bind and nucleate actin, inducing the formation of stress fibers [81].

VopL possess an N-terminal PRD, followed by three tandem repeats of WH2 motifs, with a second PRD present between the second and third WH2 motifs, and ends with a VopL C-terminal domain (VCD) that mediates dimerization (Figure 1F). Despite the presence of a tandem arrangement of three WH2 motifs, structural and biophysical studies have revealed intriguing features of how VopL is able to nucleate, but not elongate, actin assembly de novo from the pointed end [82,83,84,85].

Pyrene-actin assembly and TIRF microscopy experiments demonstrate that all three WH2 motifs are required for nucleation [82,83], and that profilin is not required [83]. However, the nucleation activity of constructs containing the three WH2 motifs but lacking the VCD is weak compared to actin alone. This confirms that, as demonstrated for eukaryotic actin nucleators that utilize a tandem repeat of WH2 motifs, their mere presence and recruitment of actin monomers per se is not sufficient to drive efficient nucleation, or rather, stabilize a small number of actin monomers in a conformation that is filament-like [49]. Both spire [86,87] and Cobl [88,89,90] achieve efficient nucleation by indirect oligomerization via interaction with dimeric proteins: formins and BAR domain proteins, respectively. On the other hand, VopL possesses a unique all-helical VCD that mediates dimerization of VopL through a coiled-coil and contributes to nucleation by direct interactions with the pointed end of the recruited actin monomers [82,83,84]. Thus, it can arrange the actin monomers recruited by the WH2 motifs into an orientation that is filament-like.

Interestingly, two studies on VopF, a closely related homolog of VopL from *Vibrio cholera* that shares a similar domain organization, have proposed an alternative model whereby VopF assembles actin and associates exclusively with the barbed end, competing with capping protein. These conclusions were based on single-color TIRF, pyrene-actin assembly assays, and small-angle X-ray scattering [91,92]. A later study using three-color TIRF with both VopF and VopL [85] provided evidence that strongly disfavors this alternative model, and demonstrated that both VopF and VopL nucleate actin from the pointed end, as previously proposed [82,83,84]. Formin or capping protein, and VopL/VopF, clearly bound to the opposite ends of an actin filament [85]. These data, taken together, suggest a model where VopL initiates the formation of new actin filaments, from the pointed end, and remains briefly associated with the pointed end before dissociation (Figure 2C). VopL does not promote elongation or remain processively associated with a filament end. The role of the PRDs is unclear in VopL. Profilin and profilin-actin are rather inhibitory, and not involved in binding the PRDs to deliver actin subunits to the WH2 motifs [83,85]. Furthermore, mechanistically, it is not yet clear what stoichiometry of VopL-to-actin is necessary or sufficient to mediate nucleation, or how the binding of actin monomers to the WH2 motifs and/or the VCD mediate the conformational change of the VCD observed in the crystal structure [83,84].

### 3.4. Other Effectors

*Chlamydia* spp. use the effector translocated actin recruitment protein (TarP) to remodel actin during invasion, life within the host, and egress [93]. Whilst TarP performs multiple functions [94], including inducing actin polymerization indirectly via Arp2/3 [95], it is able to directly nucleate actin via a mechanism involving WH2 or GAB motifs and oligomerization via a central PRD [96,97]. Similarly, Salmonella invasion protein C (SipC), an actin nucleator from *Salmonella enterica* serovar Typhirium required for invasion [98], assembles actin filaments by a combination of actin-binding and oligomerization [99,100]. Vacuolar protein sorting inhibitor protein A (VipA), encoded by *Legionella pneumophila* and important for vacuolar escape, has also been shown to nucleate actin *in vitro*, despite the lack of a clear putative WH2 or GAB motif, though it does contain a PRD [101,102]. Finally, a recent study explored the intrinsically disordered XopR from the plant pathogen spp. *Xanthomonas*, demonstrating multiple levels of manipulating actin upon infection, including its ability to directly assemble actin filaments using a WH2-like motif and self-association under different ionic strength conditions [103].

## 4. Lessons Learned

A common element found in actin assembly factors, both pathogenic and eukaryotic, is the presence of actin binding motifs, such as the WH2 motif (VopL) and the related GAB or FAB (BimA) (Figure 3). Both these effectors have an architecture whereby these motifs are connected by flexible linkers, a feature that has also been observed for eukaryotic actin nucleators [23,24]. Though Sca2 does not harbor canonical actin binding motifs identifiable by sequence (the assumption of it having predicted WH2 motifs [54,55] has been disproven [59]), it has evolved an all-helical fold that enables it to bind at least two actin subunits [59], which is consistent with structures of actin-binding proteins almost exclusively using helical folds to bind actin [104]. It should be noted that it is worthwhile to experimentally elucidate actin-binding properties of these effectors, since slight differences in the WH2 or WH2-like actin binding motif can make all the difference between non-binding, G-actin-binding, or F-actin-binding. Furthermore, as seen with Sca2, it was straightforward to discern the lack of canonical WH2 motifs, and actin-binding is achieved using the entire passenger domain [59].

Another common feature is the presence of PRDs. In eukaryotic actin nucleators, PRDs usually occur N-terminally to any actin-binding motif and are involved in either binding specialized protein domains such as SH3 domains or profilin-actin [105,106]. Whether they play a role in actin assembly by VopL and BimA remains to be seen, but, in the case of Sca2, the two PRDs are both important and cooperate for recruitment of actin monomers from profilin-actin for processive barbed-end elongation. Again, experimental elucidation has clarified their role, suggesting a model whereby they are both positioned close to the barbed end in an orientation that is optimal for actin addition at the barbed end from profilin–actin complexes. This is in contrast to formins, which elongate from profilin-actin by flexibly connected FH1 domains harboring multiple PRDs [33].

Finally, the recruited actins must be physically oriented in a manner that is sufficient to stabilize small oligomers (dimers, trimers, or tetramers), in order to facilitate the addition of further actin subunits [23,24,25,26]. Here, oligomerization is known to play an important role, as seen for eukaryotic actin assembly factors: formins dimerize via their FH2 domains, Ena/VASP tetramerizes through a C-terminal coiled-coil motif, and tandem monomer-binding nucleators dimerize indirectly, by associating with dimeric proteins, for efficient actin assembly [24]. VopL uses a unique VCD that both dimerizes and makes direct contacts to stabilize and arrange recruited actin monomers. BimA trimerization via its C-terminal coiled-coil, along with flexibility between this region and its actin-binding motifs, are both presumably important and contribute to orienting recruited actin monomers. Sca2 does not oligomerize but appears to use the entire monomeric passenger domain to encircle, recruit, and stabilize two (or more) actin monomers.

Thus, while many of the features found in eukaryotic actin assembly factors appear to be shared, the pathogen effectors discussed here adopt a minimalist approach and display a different variety of domain architectures, domains, folds, and mechanisms to functionally mimic their eukaryotic counterparts [16]. Sca2 has evolved a novel fold and actin-binding motifs to bind and encircle two actin monomers in a manner analogous to formins, with its PRDs situated within the passenger domain. BimA makes use of actin-binding motifs, along with C-terminal trimerization, and, being an autotransporter, is likely clustered on the membrane of the bacterium. VopL uses tandem WH2 motifs, coupled with its unique VCD that mediates dimerization and makes important contacts with the recruited actin monomers to arrange them into a polymerization-competent nucleus.

## 5. Outlook and Perspective

The study of pathogen effectors, their interactions with the infected host cell macromolecules, and their impact on host cell biology, has generally yielded major insights into both the biology of the pathogen and the infected host [107,108]. This review focuses on the insights gained from the study of a class of pathogen effectors that directly assemble actin filaments on our general understanding of the basic principles of actin assembly at a structural and molecular level, which has been [23,24], and continues to be [25,26], of immense interest. These effectors display some similarities with eukaryotic actin assembly factors, but, as discussed earlier, they have evolved novel mechanisms.

Although these effectors are able to directly assemble actin filaments, it does not exclude the possibility of their performing other functions, or requiring other host factors for pathogenicity, an area which should be explored. Finally, orthologues of the effectors discussed here, belonging to different phylogenetic groups of the same species, are often different in sequence and mechanism of actin assembly [7,16]. In *Rickettsia*, the Sca2 passenger domain from ancestral, transitional, and typhus groups is completely different from the spotted-fever group Sca2 described here, and they all form actin comet tails, albeit with different shape and frequency [7]. In *Burkholderia*, the BimA passenger domain from *Burkholderia thailandensis* differs from the *Burkholderia pseudomallei* BimA described here; *Burkholderia thailandensis* forms short and curved actin tails as a result of its BimA hijacking host Arp2/3 complex, instead of directly assembling actin filaments [7]. Recent developments in microscopic methods should enable significant advances in our understanding of the mechanisms employed by these pathogen effectors to assemble actin, including ones that are yet to be discovered and studied, directly benefiting our understanding of both the pathogen and actin assembly by the infected host—*Homo sapiens*.

## Figures and Tables

**Figure 1 ijms-23-11606-f001:**
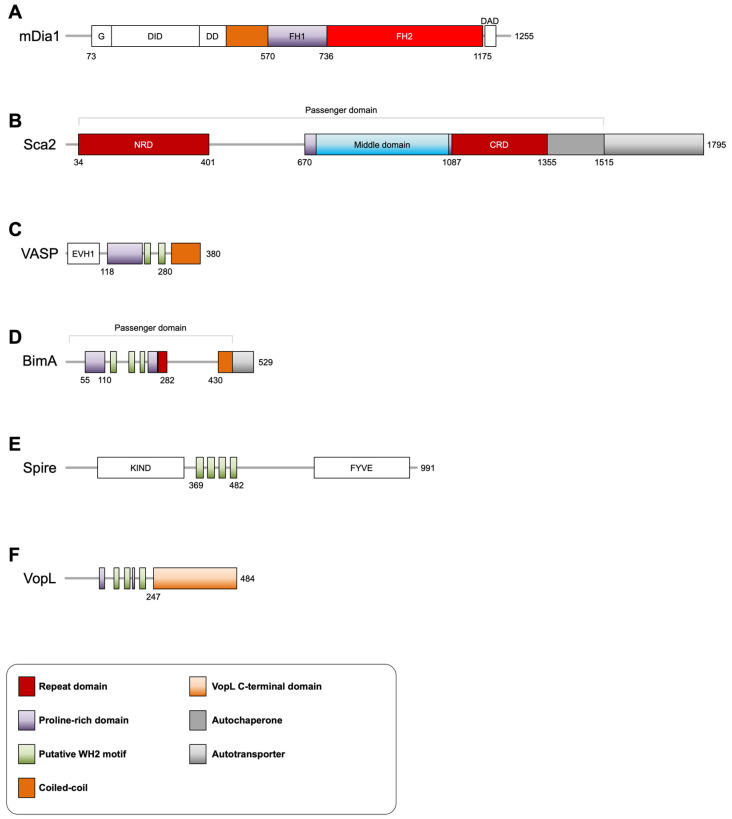
Domain organization of eukaryotic actin assembly factors and their pathogen effector mimics. (**A**) The domain organization of the formin mDia1 (mouse, Uniprot O08808) and (**B**) its pathogen mimic *Rickettsia conorii* Sca2 (Uniprot Q92JF7). (**C**) The domain organization of VASP (human, Uniprot P50552) and (**D**) its pathogen mimic *Burkholderia pseudomallei* BimA (Uniprot B9U4V1). (**E**) The domain organization of Spire (Drosophila, Uniprot Q9U1K1-1) and (**F**) its pathogen mimic *Vibrio parahaemolyticus* VopL (Uniprot Q87GE5). Domain abbreviations are as follows: G (GTPase-binding domain), DID (diaphanous inhibitory domain), DD (dimerization domain), FH1 (formin homology 1 domain), FH2 (formin homology 2 domain), DAD (diaphanous autoregulatory domain), NRD (N-terminal repeat domain), CRD (C-terminal repeat domain), EVH1 (Ena/VASP homology 1 domain), KIND (kinase non-catalytic C-lobe domain), FYVE (Fab1/YOTB/Vac1/EEA1 zinc-binding domain).

**Figure 2 ijms-23-11606-f002:**
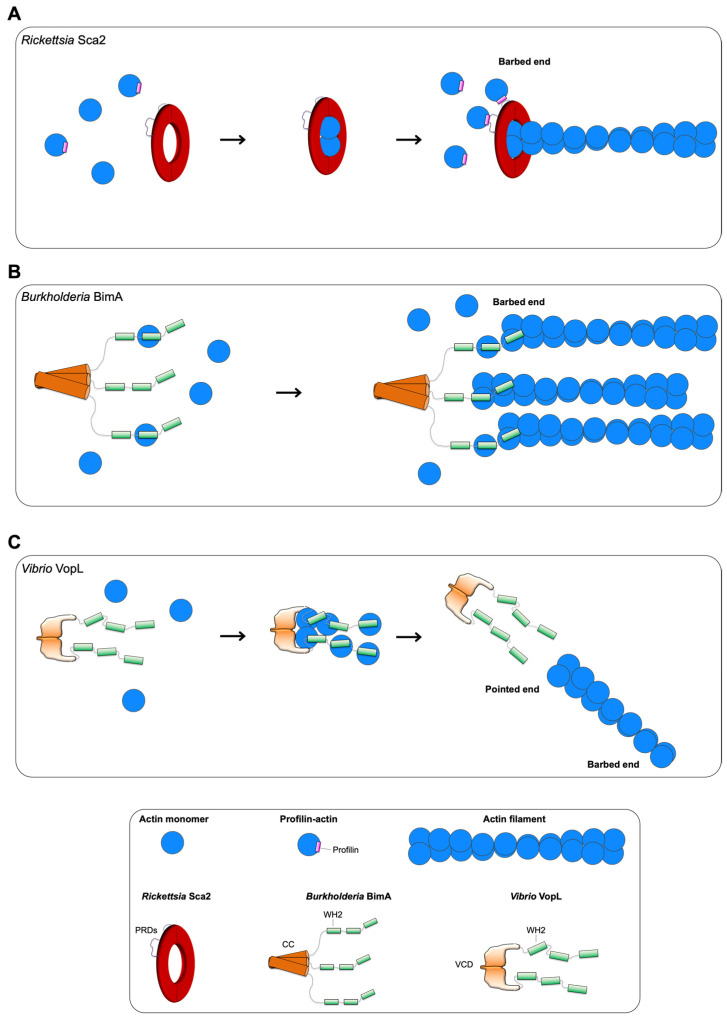
Current models of actin assembly by pathogen effectors that directly assemble actin. (**A**) Sca2 is proposed to adopt a formin-like mechanism, nucleating actin by encircling two actin subunits, and elongating the filament by tracking the barbed end and adding (proflin-)actin via its two PRDs. (**B**) BimA is proposed to adopt an Ena/VASP-like mechanism, nucleating and elongating actin filaments by a C-terminal trimerization coiled-coil and using putative WH2/GAB motifs to bind actin monomers. (**C**) VopL is a pointed end nucleator that assembles filaments by recruiting actin monomers with its WH2 motifs and using its VCD to organize the actin monomers in a filament-like arrangement.

**Figure 3 ijms-23-11606-f003:**
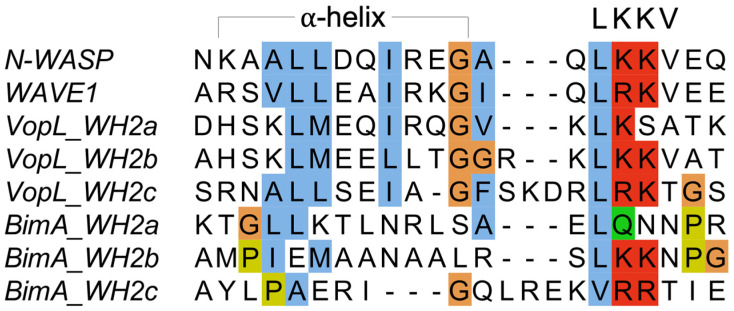
Sequence alignment of putative WH2 motifs in *Burkholderia* BimA and *Vibrio* VopL. The sequences of the known WH2 motifs of human N-WASP (Uniprot O00401) amino acids 405–424 and human WAVE1 (Uniprot Q92558) amino acids 497–516 are displayed for reference. The positions of the ⍺-helix and LKKV are shown above. Amino acid numbering for the displayed sequences is as follows: VopL WH2a 134–153, VopL WH2b 164–184, VopL WH2c 204–225, BimA WH2a 124–143, BimA WH2b 175–194, BimA WH2c 202–221.

## Data Availability

Not applicable.

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
