# Peer review of "Functional Mimicry of Eukaryotic Actin Assembly by Pathogen Effector Proteins"

_ijms, 2022, doi:10.3390/ijms231911606_

Round 1

Reviewer 1 Report

The manuscript is a review paper on the strategies utilized by various pathogenic bacteria to hijack actin of an eukaryotic host cell in order to move within the infected cell and efficiently propel itself to neighboring cells. In the Introduction, after presenting a general overview of strategies developed by pathogenic bacteria to manipulate the actin cytoskeleton, the Author narrows the scope of the review to effectors encoded by three species: Rickettsia sp., Burkholderia sp., and Vibrio sp., which express proteins mimicking functions of eukaryotic proteins that directly nucleate actin polymerization to form unbranched filaments.

The manuscript is well organized and focused, the topic is well described and illustrated. In my opinion, it sums up the current knowledge on the subject and shows the gaps that require further research.

Overall, the manuscript is well written, but has a few minor issues that need to be fixed.

 1. ‘tropomodulin’ should be removed from the sentence in line 116, because tropomodulin only inhibits elongation at the pointed end and it has no nucleating activity.

2. The term ‘passenger domain’ should be explained because not all readers will be familiar with autotransporter proteins.

3. The text would be more comprehensive for readers who are unfamiliar with pyrene-actin assembly assay if the Author elaborated on the concept of this assay (par 3.2).

4. The expression “truncations of the N-terminal half fail to interact” (line 163-164) should be modified, because truncations themselves do not fail to interact with binding partners, they change protein structure so that it fails to interact.

5.  Names of bacteria, the words in vivo and in vitro should be in italic.

6. ‘thymosin-b4’ (line 60) should be ‘thymosin-b4’.

7. Abbreviations used in Figure 1 should be also explained in the figure caption.

8. Something really bad happened with the reference list; all spaces between words have disappeared.

Reviewer 2 Report

This a review manuscript which describes the structural and functional similarities or differences of a number of bacterial proteins to eukaryotic actin polymerisation mucleating proteins. The author gives a detailed comparison of the bacterial proteins to their eukryotic counterparts and describes the functional differences very clearly. I believe however that the manuscript could be considerably improved if the author included some graphs comparing the effects of the eukaryotic and the corresponding bacterial pathogenic proteins on for instance the actin polymerisation kinetics. Also the contribution of the described bacterial proteins to the pathogenic phenotype would be interesting provided there are already clear data by transfection studies with the pathogen proteins alone.

Minor points

1) Explain the meaning of " a gating factor" and "autotransporters".

2) lane 211:  This results in .... to remain in the cytosol (I assume there is only one cytosol).

Reviewer 3 Report

The manuscript reviews non-enzymatic bacterial effector proteins (toxins) that directly bind and promote the polymerization of actin while mimicking host assembly factors: formins, Ena/VASP, and tandem WH2-domain nucleators. The review is very well written and a pleasure to read. The language is clear, the topic is described in a logical and easy to follow manner. I have the following minor suggestions to make.

1. While VopL toxin is described, a homologous VopF toxin and the controversies in the mechanisms of affecting actin  (nucleation vs barbed end capping) is not discussed. Please add a paragraph related to this topic.

2. Similarly, it may be beneficial to enrich the review by mentioning that BimA of B. thailandensis employs pathogenic mechanisms different from those of other BimA toxins.

3. Line 57: Whether dimers and trimers are unstable is controversial. Please see a recent paper in Biophysical Journal (Rosenbloom et al., 2021).

4. Lines 58-59: please explicitly clarify that profilin and thymosin inhibit spontaneous nucleation.

5. Line 271: Please add relevant references to the remark that Spire and Cobl engage dimeric proteins.

6. Line 329: Can PRD domains directly bind actin as it follows from the text? I am not aware of that.

7. Lines 383-567: For some reason, References do not have spaces between the words.
